# Evaluation of Postoperative Analgesic Efficacy of Ultrasound-Guided Suprainguinal Fascia Iliaca Block in Knee Arthroplasty: Prospective, Randomized, Feasibility Study

**DOI:** 10.3390/jcm12186076

**Published:** 2023-09-20

**Authors:** Hale Kefeli Çelik, Serkan Tulgar, Serkan Güler, Kadem Koç, Büşra Burcu Küçükordulu, Ramazan Burak Ferli, Lokman Kehribar, Ahmet Serhat Genç, Mustafa Süren

**Affiliations:** 1Department of Anesthesiology and Reanimation, Samsun Education and Research Hospital, Barış Bulvarı No: 199, Samsun 55090, Turkey; serkantulgar.md@gmail.com (S.T.); serkan_im@hotmail.com (S.G.); kademkoc.md@gmail.com (K.K.); busraburcu93@gmail.com (B.B.K.); rburakferli@gmail.com (R.B.F.); drmustafasuren@gmail.com (M.S.); 2Department of Orthopedics and Traumatology, Samsun Education and Research Hospital, Barış Bulvarı No: 199, Samsun 55090, Turkey; lokmankehribar@gmail.com (L.K.); ahmetserhatgenc@hotmail.com (A.S.G.)

**Keywords:** analgesia, arthroplasty, knee, nerve block, pain, regional

## Abstract

Background: Total Knee Arthroplasty (TKA) is one of the most commonly performed orthopedic procedures, and patients complain of severe pain in the postoperative period. The supra-inguinal fascia iliaca block (SIFIB) works as an anteriorly applied lumbar plexus block and is frequently used in hip surgeries. In this study, we evaluated the effect of SIFIB in patients undergoing TKA under spinal anesthesia. Methods: This study is a prospective, randomized, assessor-blinded feasibility study conducted in a tertiary hospital. Eighty-six patients with ASA I-III were initially enrolled, and after exclusions, 80 patients were randomized into two equal groups (SIFIB and control groups). The standard multimodal analgesia was applied to the control group, while SIFIB was additionally applied to the block group. The study measured the morphine requirement in PCA and pain intensity using Numeric Rating Scores between the two groups. Results: the 24-h cumulative morphine consumption was lower in Group SIFIB. Although there was a decrease in NRS at rest scores in the SIFIB group during some time periods, pain was moderate, and no differences in pain scores were recorded during exercise in all patients. Conclusions: In patients undergoing TKA under spinal anesthesia, a single shot of SIFIB results in a significant reduction in the amount of morphine consumed in hours. This effect was most likely related to a decrease in pain at rest in the SIFIF group.

## 1. Introduction

Total knee arthroplasty (TKA) is widely used worldwide to increase patient mobility, reduction of joint pain, and to improve quality of life in patients with advanced gonarthrosis [1]. TKA is usually associated with severe pain during the early postoperative period [2]. This pain can be managed using neuraxial and regional techniques as well as multimodal analgesia, which includes a combination of pharmacological agents, such as nonsteroidal analgesics, opioids, and sometimes gabapentinoids, etc., along with interventions, such as periarticular injections and peripheral nerve blocks [3].

With branches coming from the sacral plexus and lumbar plexus, the knee’s innervation is quite intricate [4,5]. Surgical field innervation is provided by the femoral nerve (FN), obturator nerve (ON), sciatic nerve (SN) and lateral femoral cutaneous nerve (LFCN) (non-incisional source of pain from tissue manipulation and tourniquet). These nerves can be targeted for perioperative analgesia [6]. Infiltration techniques targeting the posterior knee capsule (IPACK) have also been defined and are the subject of research [7].

The supra-inguinal fascia iliaca block (SIFIB) is a technique that blocks all of the components of the lumbar plexus, including the femoral nerve, the latero-femoral nerve and the obturator nerve. It is thought to function in a manner that is analogous to an anterior lumbar plexus block [8]. Previous studies have successfully demonstrated the effectiveness of SIFIB in reducing postoperative pain following hip surgery [9,10,11]. In a letter to the editor, Sanllorente-Sebastián et al. reported the successful application of SIFIB in 16 patients undergoing TKA under spinal anesthesia [12]. However, no clinical study has been conducted to evaluate SIFIB for postoperative analgesia in TKA.

The primary objective of this study was to assess the impact of SIFIB on opioid consumption within the initial 24-h postoperative period among individuals subjected to TKA under the administration of spinal anesthesia. Our hypothesis was that the addition of SIFIB performed after surgery will result in a reduction of opioid consumption compared to a control group.

## 2. Materials and Methods

This randomized, controlled study was conducted in a tertiary hospital between July and December 2022 in accordance with the Declaration of Helsinki. Local ethics committee approval (OMUKAEK 2022/289) was obtained, followed by Ministry of Health approval (22-AKD-46). Prior to trial initiation, clinicaltrials.gov (NCT05450211, 11 September 2022) was prospectively registered. Figure 1 depicts the CONSORT diagram of the study, which was conducted in accordance with the relevant guidelines. All patients were included in the study after providing written informed consent both for participation and application of interventions.

This study included 18–75 year old patients with American Society of Anesthesiologists (ASA) physiological classes I-III scheduled for elective unilateral knee arthroplasty under spinal anesthesia. Patients with the presence of any of the below criteria were excluded: those who did not accept spinal anesthesia, those with failed spinal anesthesia, those who refused consent for participation or wished to withdraw after commencement of study, those with coagulopathy, liver disease, kidney disease, allergy to the local anesthetic (LA) or other drugs used in the study, those with a body mass index (BMI) > 35 kg/m^2^, a history of chronic opioid or corticosteroid use or a pathology in the area of the block (local infection, hematoma, hernia, neoplasm, etc.), those unable to use patient-controlled analgesia (PCA) and those with a psychiatric or neurologic problem that would not allow for an evaluation of pain using the Numeric Rating Score (NRS). Furthermore, patients whose surgery lasted <40 min of >120 min were also excluded, in order to keep data homogenous.

### 2.1. Randomization and Blinding

Randomization into either the block or control group occurred using the sequentially numbered, opaque, sealed envelope (SNOSE) technique after the postoperative transfer of the patient into the regional anesthesia room. Every patient was assigned a random ID that was utilized during data collection, follow-up, and analysis. The physician in charge of data collection (M.S.) was blind to the group assignment of the patient. The author (S.T.) who conducted the simple randomization also administered the block but had no role in the collection or analysis of data.

### 2.2. Application of Spinal Anesthesia and Surgical Procedure

Patients were not given premedication, and the anesthesia management and perioperative analgesia plan were identical for all patients. The patients were subjected to simple intraoperative monitoring, including electrocardiogram, noninvasive blood pressure monitoring, and peripheral oxygen saturation monitoring. All patients were seated while receiving spinal anesthesia, which was administered using a 26 G spinal needle from the midline to deliver 3 mL of 0.5% heavy bupivacaine to the L3-L4/L4-L5 region. No additive medication was added to the spinal anesthetic. Following spinal anesthesia, patients underwent pinprick tests at regular intervals, and the surgical procedure began once sensory block was confirmed to have occurred at the level of the T10 dermatome. Using the standard technique, a bimalleolar TKA was performed with a tourniquet. At the conclusion of the procedure, no additional analgesia, such as a periarticular injection, was administered by the surgeon.

### 2.3. Application of SIFIB Block

The SIFIB was performed immediately following surgery in the block performance room using an in-plane technique and a high-frequency linear transducer (10–18 MHz, Esoate MyLab™30Gold, Genoa, Italy). The transducer was positioned on the femoral crest to sono-visualize the femoral artery, femoral nerve, fascia iliaca, and iliacus muscle. By shifting the transducer laterally, the fascia iliaca continuity between the iliacus and sartorius muscles was displayed. The transducer was rotated superolaterally to the oblique plane and displaced just medial to the anterior superior spina iliaca. Sonoanatomically, the abdominal muscles, the deep circumflex artery, the iliacus muscle, the Sartorius muscle, and the fascia iliaca were identified in the same image. Using the in-plane technique, an 85 mm peripheral block needle (Vygon Echoplex, 85 mm, 21 G, Ecouen, Paris, France) was guided from the caudal to the cephalad and advanced into the space between the fascia iliaca and the iliacus muscle. After injecting 1 mL of saline to confirm the tip-of-needle position, 40 mL of 0.25% Bupivacaine was slowly injected into this area. The plane and spread were confirmed with the extension of the LA to the area just below the deep circumflex artery.

### 2.4. Standard Perioperative Analgesia Protocol and Follow-Up of Pain

A standard analgesia regimen was used in all patients. Following conclusion of surgery, 1 g iv paracetamol and 50 mg iv dexketoprofen were administered and repeated every 8 and 12 h, respectively. All patients also received intravenous PCA in the recovery room. The PCA, set at no basal infusion, contained morphine at a concentration of 0.3 mg/mL, a total volume of 100 mL, bolus dose of 1 mg, and a lock-in time of 20 min. PCA was given to the patient immediately after the block was performed, and the patient was advised to request analgesia if the NRS score was ≥4. Total opioid consumption was recorded at the 1st, 3rd, 6th, 12th, 18th and 24th postoperative hours. If during the first 24 h the NRS (numeric rating scale) score was ≥4 despite the use of PCA, 25 mg meperidine iv was administered as rescue analgesia.

Patients’ pain was recorded both at rest (NRS-S: static pain) and during motion (NRS-D: dynamic pain). NRS-S was recorded at the 1st, 3rd, 6th, 9th, 12th, 18th, and 24th postoperative hours, and the NRS-D score was recorded at the 9th, 12th, 18th, and 24th postoperative hours. The NRS is used in adults as a one-dimensional measurement of pain intensity. The patient’s pain is scaled on a numerical, eleven-point scale ranging in increasing order from no pain (0) to the worst pain imaginable (10).

### 2.5. Outcome Measurements

The primary outcome was the cumulative opioid requirement in the first 24 h. The NRS scores obtained at various time points constituted the secondary outcome. In addition to our findings, nausea, vomiting, time to first analgesia request, and quadriceps weakness were noted from the 9th postoperative hour. In the evaluation of quadriceps muscle weakness, the patient’s knee was actively extended, first against gravity and then against applied resistance, while the hip joint was flexed at 45° and the knee joint at 90°. The patient’s ability to extend the knee was evaluated and scored on a 3-point scale, as follows: 0 = Normal force (patient can extend the knee against both gravity and applied force), 1 = Paresis (patient can extend the knee by overcoming gravity but cannot overcome resistance), and 2 = Paralysis (patient cannot extend the knee).

### 2.6. Statistical Analyses

In a pilot study conducted with 20 patients randomized into two groups, the average 24-h cumulative morphine requirement was found to be 8.6 ± 7.24 mg and 14.1 ± 5.91 mg, for the SIFIB and control groups, respectively. Using this data and α = 5%, β = 10%, and power = 95%, the minimum required sample for a single group was calculated to be 38. Considering loss to follow-up or dropout, 40 patients were included.

Statistical analysis was completed using Statistical Package for Social Sciences (IBM Corp. Released 2013. IBM SPSS Statistics for Windows, Version 22.0. Armonk, NY, USA: IBM Corp.). The Kolmogorov–Smirnov test was utilized for evaluation of normality. The mean and standard deviation and/or median and interquartile range (25th and 75th percentiles) were used to present descriptive data. The Student’s t test was used to compare continuous variables, and the Chi squared test was used to compare ratios. Fisher’s exact test was used to compare categorical variables (ASA classification, gender, etc.). The first analgesia requirement time was analyzed using Kaplan–Meier analysis. The threshold for statistical significance was set at *p* < 0.05. The Bonferroni correction was used for the analysis of NRS scores, with statistical significance adjusted to *p* < 0.0071 due to measurements from 7 time points.

## 3. Results

Figure 1 depicts the study’s CONSORT diagram. Following the screening of 86 patients, 6 were excluded (2 for failed spinal anesthesia and 4 for prolonged surgical time), and randomization occurred for the remaining 80.

Demographic data, as shown in Table 1, was similar between groups. The average surgical times were similar between the SIFIB and control groups (56.12 ± 12.37 vs. 57.87 ± 14.27, *p* = 0.559). As indicated in Table 1 and Figure 2, the time to first analgesia requirement was significantly longer in the SIFIB group when compared to the control group (223.75 ± 128.52 vs. 140.25 ± 97.16, *p* = 0.001) (Table 1).

NRS-S scores were significantly less in the SIFIB group at the 3rd, 12th and 18th h, while there was no difference at the other time points. NRS-D scores at the 9th, 12th, 18th and 24th h were similar between groups. The distribution of NRS-S and NRS-D scores is shown in Table 2.

Measurements at all time points revealed that cumulative morphine consumption was statistically significantly lower in the SIFIB group at all time points when compared with the control group, as shown in Table 3. Cumulative morphine consumption at the 24th h was 7.97 ± 9.88 mg for SIFIB and 16.12 ± 7.38 mg for the control group (*p* < 0.001). In addition, the box plot graph that demonstrates the cumulative morphine requirement in time frames is presented in Figure 3.

Two patients in the control group and three in the block group experienced nausea/vomiting requiring ondansetron. While there was no motor weakness at the 9th h in the control group, 13/40 patients in the SIFIB group had grade 1 motor weakness at the 9th h and 6/40 patients had grade 1 motor weakness at the 12th h. At the 18th and 24th h, no patients in the SIFIB group experienced any motor weakness. No meperidine dose was administered as rescue analgesia.

## 4. Discussion

Our study has demonstrated that in most time frames, single-shot ultrasound-guided SIFIB applied at the end of an operation as part of multimodal analgesia in patients undergoing TKA under spinal anesthesia reduced the need for cumulative analgesia and resulted in a decrease in NRS-S scores. However, both groups’ pain levels at rest were within acceptable limits.

Due to the limitations of the traditional fascia iliaca compartment block (or 3-in-1 block), such as missing the lateral LFCN and obturator nerve, clinicians have sought alternatives for anterior blocking of the lumbar plexus components. SIFIB was defined as an ultrasound-guided technique by Hebbard in 2011 [8]. In the early years, this new technique was the subject of only a few studies. However, SIFIB has become popular in hip surgeries in recent years, owing to the widespread use of ultrasonography in regional anesthesia practice [13,14]. In our study, however, we used SIFIB for analgesia in TKA patients. In the literature, the use of SIFIB in procedures distal to the hip is limited [12,15].

Several studies have investigated the ideal volume required for SIFIB to block all components of the lumbar plexus and the ideal concentration to provide adequate analgesia. In a magnetic resonance imaging and sensorial analysis study on volunteers, Vermeylen et al. reported that SIFIB spread to the lumbar plexus was more consistent than the infrainguinal technique, and 40 mL of LA was required for blockade of the three targeted nerves [13]. In a study of 60 cadavers, the minimum effective volume (MEV 90) required for the blockade of all three FN, LFCN, and ON in a SIFIB application was reported to be 62.5 mL [16]. On the contrary, MEV 95 for ropivacaine was reported to be 26.99 mL, although sensorial blockage of ON was not evaluated [17]. In our study, we used a volume of 40 mL for SIFIB application. Higher volumes produce different results, and we predict that block features will change as the concentration and volume changes. In our SIFIB application, we used 0.25% bupivacaine, which is the most widely available and widely used LA in our country. In our clinic, we are currently conducting a clinical trial for bupivacaine, which includes sensory assessments of all three nerves to determine MEV in SIFIB (NCT05408585).

Regional analgesia techniques are commonly used in the multimodal analgesia of TKA, but their use is controversial owing to the potential negative effects on rapid recovery [18]. Due to the negative effects of neuraxial techniques and nerve blocks on rapid recovery (quadriceps weakness, nerve damage, urinary retention, and so on), clinicians have turned to techniques such as adductor duct block (ACB) and local infiltrative analgesia (LIA) [18,19,20]. In a small number of cases reported to us regarding the SIFIB application, motor weakness was observed. We believe that ‘differential block’ can be achieved by developing sensorial block without motor block using different LA volumes and concentrations. Prospective research is required on this topic.

The posterior innervation of the knee capsule is supplied by the posterior branch of the obturator nerve (ON) and sciatic nerve branches [5]. Some studies have reported the complimentary use of the IPACK block, which provides relief for postoperative pain originating from the posterior of the knee capsule [21,22]. In this study, SIFIB was exclusively applied to the lumbar plexus. In this study, we did not combine a SIFIB and IPACK block because of the large volume of local anesthetics required to perform a SIFIB. Additionally, from a methodological point of view, it is possible that the addition of an IPACK block may have complicated the interpretation of our data.

Some orthopedic surgeons apply local infiltration analgesia (LIA) to the posterior part of the knee and/or the cutaneous area before closing the surgical field, and some apply a special analgesic/antibiotic mixture to the knee, which are not routine in our country.

Although we have demonstrated that SIFIB has a statistically lower 24-h opioid requirement and lower NRS scores when compared to the control group, this does not necessarily imply that this statistical difference is clinically significant [23]. In a systematic review, the minimal clinically important differences (MCID) in TKA and total hip arthroplasty were determined to be “10 mg i.v. morphine equivalents or 40% for opioid consumption and 15–18 mm or 30 percent for pain scores” [23]. In our study, 24-h morphine consumption was reduced by more than 50%. We did not detect a significant difference in NRS scores; however, multimodal analgesia led to acceptable NRS scores in the control group. Furthermore, since the need for opioids is minimal in the first few hours after spinal anesthesia, it may be advantageous to re-evaluate the MCID values with homogenized groups.

Our study has several limitations. First, this study was designed based on volumes and doses we studied in our preliminary study. It is possible that different volumes and concentrations may have produced stronger or weaker results. Studies that investigate the quality of recovery will be more clinically relevant than those that evaluate the efficacy of regional anesthesia techniques based solely on opioid requirement or pain scores. When we planned this study, the Turkish validation of QoR-15 had not been published, so we were unable to utilize it [24]. In our study, quadriceps weakness was measured using a simple scale. In addition, studies should be conducted to determine the effects of anesthesia on quadriceps strength, function, and recovery using tests such as the “timed up and go test”, “quadriceps muscle strength”, “joint range of motion test”, and the ‘five time sit to stand test”. This will serve as a guide for the fields of orthopedics and physical therapy. Nowadays, early recovery guidelines, which are defined and recommended for surgical procedures, have an important place in the literature [25].

LIA was not a part of our protocol. It is possible that from a functional point of view, the addition of LIA would have been more beneficial. Therefore, our study is a feasibility study focused on pain, and the clinical implications for SIFIB applied in TKA should be studied separately.

## 5. Conclusions

Our investigation has shown that a single-shot SIFIB in patients undergoing TKA under spinal anesthesia results in a significant reduction in the amount of morphine consumed in the first 24 h after surgery as well as an improvement in the quality of postoperative analgesia at rest, but not during mobilization.

## Figures and Tables

**Figure 1 jcm-12-06076-f001:**
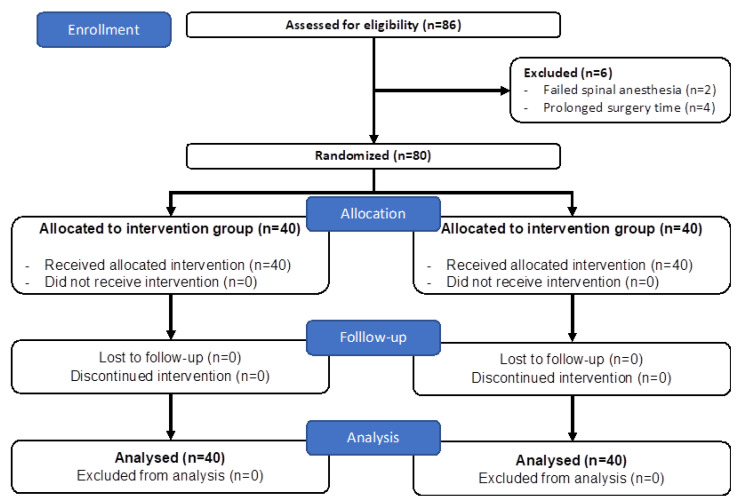
CONSORT diagram of the study.

**Figure 2 jcm-12-06076-f002:**
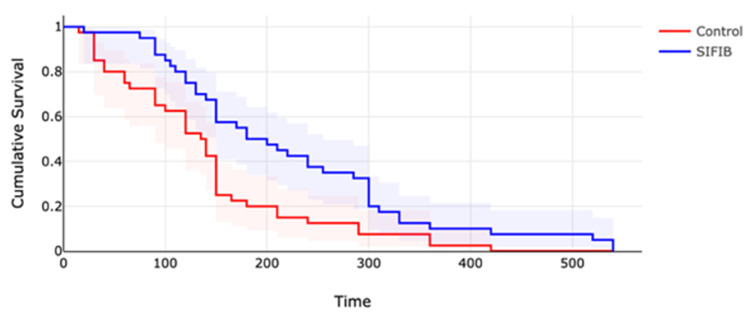
Kaplan Meier curve for SIFIB and control group with confidence intervals.

**Figure 3 jcm-12-06076-f003:**
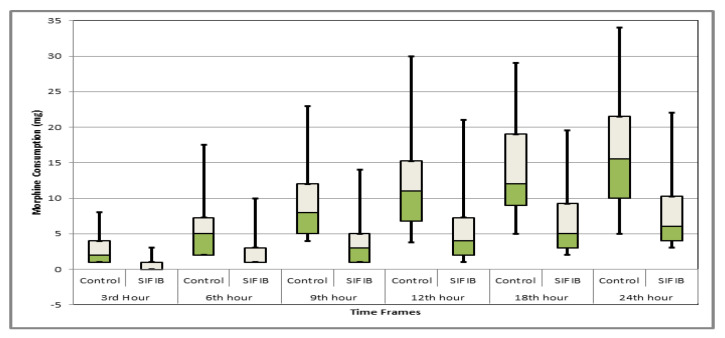
Box plot comparison of morphine consumption at different time frames between groups.

**Table 1 jcm-12-06076-t001:** Comparison of age, gender, ASA classifications, height, weight, body mass index, surgical time and time to first analgesia requirement between groups.

	SIFIB (n = 40)	Control (n = 40)	*p*
Age (years)	64.45 ± 6.77	65.45 ± 7.15	0.522
Gender (Female/Male)	36/4	38/2	0.395
ASA Classification (I/II/III)	2/31/7	2/34/4	0.619
Height (cm)	162.45 ± 6.34	160.22 ± 4.75	0.078
Weight (kg)	81.72 ± 8.74	78.85 ± 7.60	0.121
BMI (kg/m^2^)	31.0 ± 3.46	30.69 ± 2.47	0.645
Surgical Time (mins)	56.12 ± 12.37	57.87 ± 14.27	0.559
Time to First Analgesia Requirement (mins)	223.75 ± 128.52	140.25 ± 97.16	**0.001**

The data are presented as mean ± standard deviation or as a number. *p* values indicate statistical significance and bolded.

**Table 2 jcm-12-06076-t002:** Comparison of NRS scores at rest (NRS-S) and during movement (NRS-D) at different time points between SIFIB and control groups.

NRS-S	SIFIB (n = 40)	Control (n = 40)	*p*
1st h	1 (0.75–2)	2 (0.75–2)	0.097
3rd h	2 (1–2)	3 (2–3)	**<0.001**
6th h	2.5 (2–3)	3 (2.75–3)	**0.001**
9th h	2 (2–3)	3 (2–3)	**0.019**
12th h	2 (1–3)	3 (2–3)	**0.002**
18th h	2 (2–3)	2 (2–3)	**0.001**
24th h	2 (2–3)	2 (1–3)	0.050
**NRS-D**			
9th h	3 (3–4)	4 (3–4)	0.288
12th h	3 (3–4)	3 (3–4)	0.219
18th h	3 (2–4)	3.5 (2–4)	0.101
24th h	3 (2–3)	3 (2–4)	0.278

The data are expressed as the median (percentiles 25–75). *p* values in bold indicate statistical significance and bolded. h: hours.

**Table 3 jcm-12-06076-t003:** Comparison of cumulative morphine consumption at different time points between SIFIB and control groups.

Cumulative Morphine Consumption	SIFIB (n = 40)	Control (n = 40)	*p*
1st h	0 (0–0)	0 (0–0.25)	**0.032**
3rd h	0 (0–1)	2 (1–4)	**<0.001**
6th h	1 (1–3)	5 (2–7.25)	**<0.001**
9th h	3 (1–5)	8 (5–12)	**<0.001**
12th h	4 (2–7.25)	11 (6.75 ± 15.25)	**<0.001**
18th h	5 (4–10.25)	12 (9–20)	**<0.001**
24th h	6 (4–10.25)	15.5 (10–20.5)	**<0.001**

Data are expressed as median (percentiles 25–75). *p* values that are written in bold represent statistical significance. h: hours.

## Data Availability

Data are available upon request from the corresponding author.

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
