# Peer review of "Evaluation of Postoperative Analgesic Efficacy of Ultrasound-Guided Suprainguinal Fascia Iliaca Block in Knee Arthroplasty: Prospective, Randomized, Feasibility Study"

_jcm, 2023, doi:10.3390/jcm12186076_

Round 1

Reviewer 1 Report

Nice performed and written study.

Few suggestions to consider:

- Influence of tournique on pain?

- LIA is quite common in TKA. Why not used?

- How did the motorical weakness influence early recovery (f.i. out of bed after 2 to 6 hours?

- why no pre-medication is given?

- Are there any data on pain after first 24h? Would be very interesting to read about the pain on day 2,3 and 4. How does the pain develop?

- Although mentioned in the discussion, I see no references, nor really link to Fast Track/Repid recovery.

These are suggestion to consider that could improve the paper.

Author Response

- Influence of tournique on pain?

In our study, TKA was performed under spinal anesthesia, and there was no such peroperative complaint. We did not evaluate tourniquet-related pain in the postoperative period.

- LIA is quite common in TKA. Why not used?

We did not do it in LIA in our study, so that the effect of SIFIB would not be overshadowed and block coverage areas would not overlap.

- How did the motorical weakness influence early recovery (f.i. out of bed after 2 to 6 hours?

Unfortunately, we did not evaluate the effects on motor weakness and early recovery in this time frame. And we stated this issue as a limitation.

- Why no pre-medication is given?

Since the patient age group is relatively high, we avoided premedication unless necessary.

- Are there any data on pain after first 24h? Would be very interesting to read about the pain on day 2,3 and 4. How does the pain develop?

Unfortunately, we do not have any data on pain after the first 24 hours. We plan to evaluate early (first 6 hours) and late (up to 72 hours) pain in our study, whose planning and ethical approval process is ongoing.

- Although mentioned in the discussion, I see no references, nor really link to Fast Track/Repid recovery.

Reference added to manuscript.

These are suggestion to consider that could improve the paper.

Thanks for your suggestions, and we agree with you for improving our manuscript.

Reviewer 2 Report

Dear Authors.

After reviewing the document, I have to make the following comments and considerations:

- The abstract seems to me adequate, although its different sections should be specified as indicated by the journal's standards: Background, Methods, Results and Conclusions.

- The introduction of this article is well elaborated, briefly describing the existing problems of the research topic.

- The objective should be stated in a more specific way since the aim is to study the impact of this blockade on the use of opioids for postoperative analgesia in these patients.

- The methodology is clearly and adequately described, allowing the study problem to be addressed and the proposed objective to be achieved. The randomization technique used should be explained more clearly because the relationship between sealed envelope and simple randomization is not adequately clear.

- The results have been presented in a clear manner to facilitate their understanding.

- The discussion provides a detailed and in-depth analysis of the results obtained, establishing relationships between the findings and previous studies in the scientific literature.

- The conclusion of this research is consistent with the results obtained and adequately responds to the proposed objective.

- The references are appropriate to address this topic of study, although they should be adapted to the standards of the journal.

Kind regards.

Author Response

- The abstract seems to me adequate, although its different sections should be specified as indicated by the journal's standards: Background, Methods, Results and Conclusions.

Many thanks for your kind words. We arranged our abstract as you stated.

- The introduction of this article is well elaborated, briefly describing the existing problems of the research topic.

Many thanks for your kind words…

- The objective should be stated in a more specific way since the aim is to study the impact of this blockade on the use of opioids for postoperative analgesia in these patients.

We clearly stated our objective as ‘The primary objective of this study was to empirically examine the impact of SIFIB on the opioid consumption within the initial 24-hour postoperative period among individuals subjected to TKA under the administration of spinal anesthesia. To this end, our study sought to discern whether the utilization of SIFIB resulted in a notable reduction in opioid requirements compared to a control group.’

- The methodology is clearly and adequately described, allowing the study problem to be addressed and the proposed objective to be achieved. The randomization technique used should be explained more clearly because the relationship between sealed envelope and simple randomization is not adequately clear.

Changed to ‘Sequentially numbered, opaque, sealed envelope (SNOSE) technique’

- The results have been presented in a clear manner to facilitate their understanding.

- The discussion provides a detailed and in-depth analysis of the results obtained, establishing relationships between the findings and previous studies in the scientific literature.

- The conclusion of this research is consistent with the results obtained and adequately responds to the proposed objective.

- The references are appropriate to address this topic of study, although they should be adapted to the standards of the journal. Corrected

Many thanks for your kindly words….